# The Next Step for MRD in Myeloma? Treating MRD Relapse after First Line Treatment in the REMNANT Study

**Anne-Marie Rasmussen [1,2], Frida Bugge Askeland [1,2] and Fredrik Schjesvold [1,2,*]**

1   Department of Hematology, Oslo Myeloma Center, Oslo University Hospital, 0450 Oslo, Norway; annemra55@gmail.com (A.-M.R.); friask@ous-hf.no (F.B.A.)
2   KG Jebsen Center for B Cell Malignancies, University of Oslo, 0450 Oslo, Norway
*   Correspondence: fredrikschjesvold@gmail.com; Tel.: +47-996-97-796

**Abstract:** The treatment approach for multiple myeloma (MM) has changed in recent years. After the approval of maintenance treatment after stem cell transplant in younger patients, the paradigm of continuous treatment is now prevailing in all clinical situations of myeloma. However, the best time to initiate relapse treatment is still unclear. With increased frequency of minimal residual disease (MRD) negativity, and the established clinical benefit of this finding, one of the large clinical questions in myeloma is how to approach MRD re-appearance. In this paper, we go through the MRD technology, existing and possible uses of MRD in the clinic, and data for early treatment before we introduce the design of the ongoing REMNANT study; a randomized study with early treatment of MRD relapse after first line treatment.

**Keywords:** MRD; VRd; multiple myeloma; early relapse treatment

## 1. Introduction

Myeloma is a neoplastic expansion of bone marrow (BM) plasma cells, and typical clinical features are anemia, renal failure, and breakdown of skeletal bone with accompanying pathological fractures and bone pain. The disease is incurable, and most patients experience a continuous cycle of treatment remissions and relapses, frequently with more than 10 treatment lines before the patients succumb to the disease. The best and longest responses are achieved in the first lines of treatment.

Outcomes for patients with multiple myeloma (MM) have improved substantially during the last decades, leading to improvement in quality of life (QoL) and overall survival (OS). The availability of several new drugs and drug combinations (proteasome inhibitors, immunomodulators, and monoclonal antibodies) has produced deeper and more long-lasting responses, both in younger and elderly MM patients. In 2014, new sub-clinical myeloma-defining biomarkers were included in the definition of MM, which provide earlier treatment for many patients and have been introduced in some centers. Treatment of high-risk smoldering myeloma (HR-SMM) is on the rise, is being explored in many clinical studies, and has even been recommended by some experts in the field. Changes in treatment paradigm may have the potential to significantly limit disease-related complications such as fractures and kidney failure and improve QoL for patients with MM.

Until recently, achievement of both sustained complete response (CR) and stringent CR was the treatment goal and was associated with improved progression free survival (PFS) and OS. However, most patients in >CR eventually relapsed, indicating that resistant sub-clones were present, even after intensified therapy. This demonstrates the need for more sensitive methods to monitor minimal residual disease (MRD), and new technologies have provided the way to analyze this.

## 1.1. MRD Negativity Increase PFS and OS in NDMM and RRMM

There is now solid evidence that bone marrow (BM) based MRD assessment is one of the strongest prognostic factors in myeloma, and deeper responses correlate with favorable outcomes.

In the IFM 2009 trial [1], 700 newly diagnosed MM (NDMM) patients were treated with either eight cycles of bortezomib, lenalidomide, and dexamethasone (VRd) or three cycles of VRd followed by autologous stem cell transplant (ASCT) and two cycles of VRd consolidation. Both arms received lenalidomide maintenance for one year. In order to compare whether the level of sensitivity affected outcome, MRD analysis was performed with next generation flow-cytometry (NGF) and next generation sequencing (NGS). They demonstrated that an MRD level below $10^{-6}$ is predictive of superior PFS compared with $10^{-5}$ or $10^{-4}$. Patients who were MRD negative had a higher probability of prolonged progression-free survival than patients with detectable residual disease, regardless of treatment group (VRd vs. transplant), cytogenetic risk profile, or International Staging System disease stage at diagnosis.

Three meta-analyses [2–4] have demonstrated the role of MRD status in relation to clinical outcome in newly diagnosed MM (NDMM) patients. The paper by Munshi et al. [4] demonstrated that achieving MRD negativity in NDMM patients was associated with significantly better PFS and OS (see Figure 1) and that MRD was the better predictor of PFS and OS compared to CR. Findings from this meta-analysis provide evidence to support the basis for integrating MRD assessment into the management of MM.

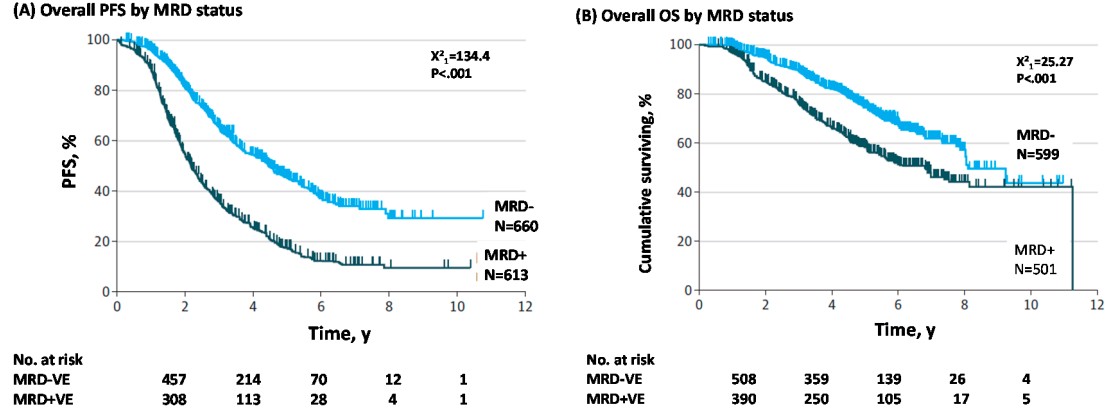

**Figure 1.** Kaplan-Meier curves for progression free survival (PFS) (**A**) and overall survival (OS) (**B**) comparing minimal residual disease (MRD)-negative versus MRD-positive myeloma patients. Data were adjusted to account for the different proportions of patients in each study being MRD-positive and MRD-negative. Reproduced with permission from Munshi et al. [4] Association of Minimal Residual Disease with Superior Survival Outcomes in Patients with Multiple Myeloma: A Meta-analysis. Copyright 2017 American Medical Association. All rights reserved.

The meta-analysis by Landgren et al. [3], including four studies of NDMM patients, concluded that, although inherent differences across the studies were present (drug use and MRD assay sensitivity), all Hazard Ratios (HR) favored MRD negativity for longer PFS. The meta-analysis demonstrated that remaining MRD positive was associated with worse progression-free survival (HR = 2.85; 95% CI 2.17–3.74; $p < 0.001$). Only two of the studies provided HR for OS, and meta-analysis of the two studies showed that retaining MRD positivity was associated with a higher risk of death.

The meta-analysis of six randomized studies in NDMM patients by Avet-Loiseau et al. [2] discussed the possibility of using MRD status as a surrogate endpoint in clinical trials. In order for an endpoint to be considered a surrogate it has to predict clinical benefit, and two key criteria have to be met: [1] to, at a patient level, be correlated with the clinical benefit endpoint independent of treatment and [2] the treatment effect on the surrogate endpoint must predict the treatment effect on the clinical benefit endpoint [5]. Randomized studies that monitor treatment effect on both MRD and PFS are required to fulfill the second criterion.

The analysis revealed a clear association between MRD negativity and improved PFS and supports the use of MRD negativity as a surrogate endpoint in clinical trials. However, OS data were immature, and patients have to be followed up for documentation of PFS and OS to verify that MRD is a valid endpoint.

Until recently, there were limited data on the prognostic value of MRD in the relapsed and refractory MM (RRMM) setting. After introduction of highly effective novel drug combinations, more data have been published on correlation of MRD and outcome. A paper from 2015 by Paiva et al. [6] showed that MRD negativity after salvage therapy was associated with prolonged PFS in RRMM patients. Several recent studies with novel drug combinations have demonstrated that achieving MRD negativity in RRMM patients correlates with improved PFS [7,8].

The CASTOR study [7] was the first randomized phase III clinical trial in RRMM patients with prospective MRD evaluation. Patients were randomized to receive daratumumab, bortezomib, and dexamethasone (DVd) or bortezomib and dexamethasone (Vd). Deeper responses to DVd were associated with significantly higher MRD negative rates versus Vd. The benefit of DVd was also maintained in patients regardless of cytogenetic risk, while Vd was not able to induce MRD negativity in high-risk patients (see Figure 2). The MRD negative rate was 11.6% in the DVd arm versus 2.4% in the Vd arm. This correlated with a longer median PFS in the DVd arm (19.4 months) compared to 7.1 months in the Vd arm. OS data was not available at time of analysis. It was further demonstrated that patients achieving MRD negativity showed a longer PFS, independent of treatment arm.

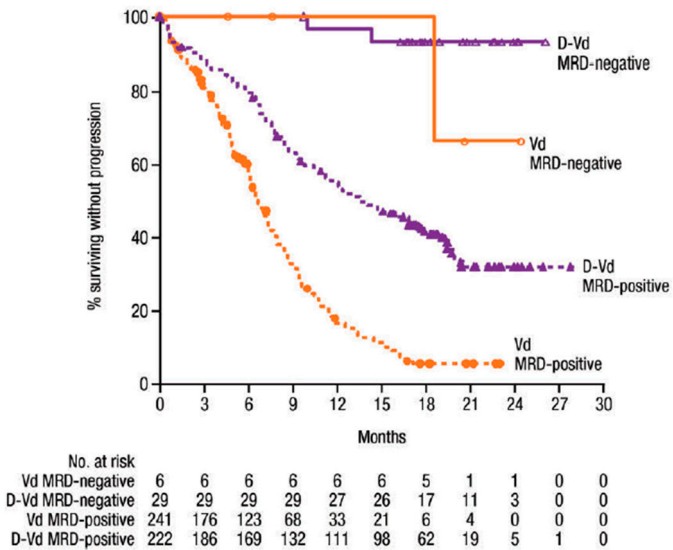

**Figure 2.** Kaplan-Meier estimates of PFS among patients evaluated for cytogenetic risk. Patients were treated with daratumumab, bortezomib, and dexamethasone (D-Vd) or with bortezomib and dexamethasone (Vd). MRD was evaluated at a sensitivity threshold of $10^{-5}$ in bone marrow (BM) aspirates using the cloneSEQ assay. Reproduced with permission from Spencer et al. [7]. Daratumumab plus bortezomib and dexamethasone versus bortezomib and dexamethasone in relapsed or refractory multiple myeloma: updated analysis of CASTOR. Copyright 2017 American Medical Association. All rights reserved.

The POLLUX study [8] randomized RRMM patients to daratumumab, lenalidomide (R), and dexamethasone (DRd) or lenalidomide and dexamethasone (Rd). The importance of achieving deep response was reflected in PFS, which was prolonged in patients who achieved MRD negativity compared with MRD-positive patients in both treatment arms. However, the number of patients who achieved MRD negativity was higher in patients who were treated with DRd. After 25.4 months of follow up, median PFS was not reached for DRd versus 17.5 months for Rd alone.

## 1.2. MRD as a Surrogate Endpoint in Clinical Trials

The time to achieve mature PFS data in future randomized trials investigating modern combination therapies in newly diagnosed myeloma patients (NDMM) is estimated to be more than seven years [9]. If PFS and OS would be the only regulatory endpoints for approval of novel drugs, it will result in unacceptable delays in delivering promising therapies to patients. There is a high need for surrogate endpoints for survival in clinical trials to determine the efficacy of novel therapies within an acceptable timeframe. MRD testing offers a unique opportunity to identify effective drugs early and give patients rapid access to new and efficacious drugs. A robust association between depth of response measured by MRD and PFS has been demonstrated in several randomized trials in NDMM and RRMM and should support the use of MRD assessment as a surrogate endpoint for all future trials in MM.

The FDA states that submissions for drug approval that use MRD for regulatory purposes or for critical treatment purposes should include sufficient information that MRD is a clinically valid biomarker for the disease and type of therapy.

## 2. Techniques for MRD Monitoring

### 2.1. How to Monitor MRD

Based on data from several studies in MM achieving MRD negativity has been associated with a better outcome and it has become clear that sensitive assays for MRD assessment in MM patients are needed. The sensitive assays for detecting MRD in BM that have been developed include cell-based NGF (EuroFlow) and molecular-based assays such as NGS.

MRD assessment has already been implemented in most phase III clinical trials in MM, and the International Myeloma Working Group (IMWG) has defined a revised criterion of response assessments for patients with MM, including MRD evaluation. The IMWG criteria for measuring MRD includes two methods for detection of MRD in BM; Euroflow and NGS, both with a recommended minimum sensitivity of 1 in $10^5$ nucleated cells or higher. Because sustained MRD negativity has been shown to correlate with better outcome [4], IMWG has implemented the assessment of long-term MRD negativity in the BM using NGF or NGS, or both, and by imaging (positron-emission tomography computed tomography (PET-CT)), confirmed a minimum of 1 year apart.

It is recommended to perform MRD testing in all patients who achieve either CR or very good partial remission (VGPR) [10]. This recommendation is based on the experience that many patients who are in VGPR are in fact negative for MRD in the BM. In some patients, this is due to the long half-life of serum immunoglobulin, which may take several months, even after virtually all tumor cells have been eradicated. This is particularly seen for the IgG isotype [10]. However, the IMWG definition of MRD negativity includes CR.

The Euroflow-based NGF method has now been established as the standardized method for MRD detection in MM [11] and has been implemented as the most convenient method for MRD assessment in clinical trials.

The method uses an eight-color panel for 12 different markers (CD38, CD138, CD45, CD19, CD27, CD28, CD56, CD81, CD117, Ig-kappa and Ig-lambda, and β2-microglobulin). Ideally, the Euroflow assay should be performed within few hours from BM aspirates because CD138 tends to be internalized by plasma cells (PCs). This can be a problem if samples have to be shipped overnight.

MRD Euroflow has a stable sensitivity of 20 cells in 10 mill ($2 \times 10^{-6}$), while a higher sensitivity can be obtained with NGS where the lower limit of detection is 10 cells in 10 mill ($1 \times 10^{-6}$), however not routinely achieved in all samples. Both Euroflow and NGS MRD assessments have been standardized using BM samples. Assessments to detect MRD in peripheral blood have been proposed by searching for circulating tumor cells (CTCs) by NGF or circulating free DNA (cfDNA) by NGS, but the sensitivity is suboptimal and current methods do not give reproducible results. In addition, some technical issues have to be solved before implementation in clinical trials [12].

There are several advantages of using Euroflow in MRD detection. The method shows high-applicability, rapid turnaround time, intrinsic quality control checks, a lack of requirement of a patient baseline sample (differently from NGS), and cost-effectiveness. The caveats for its applicability in clinic are, however, the need for experienced flow doctors, the requirement for fresh BM samples, and lower sensitivity compared to NGS.

Only a few companies can offer MRD NGS assessment with a sensitivity of $10^{-6}$. The clonoSEQ Assay developed by Adaptive Biotechnologies is an NGS-based assay that identifies rearranged IgH (VDJ), IgH (DJ), IgK, and IgL receptor gene sequences, as well as translocated BCL1/IgH (J) and BCL2/IgH (J) sequences using PCR. The NGS technology allows the processing of millions of sequence-reads in parallel, making this applicable for MRD detection. The assay requires baseline samples from patients in order to identify one or more dominant sequence(s), which will be used to track MRD in samples collected after treatment. The major advantage with molecular methods is that they can be performed in batches on stored samples. On the negative side is the cost and the requirement of baseline samples. For a summary of pros and cons of the different methods for MRD assessment in MM, see Table 1 below.

**Table 1.** Shows a comparison of Euroflow, next generation sequencing (NGS), and MALDI ToF Mass spectrometry used for MRD assessment of multiple myeloma (MM).

|  | **Euroflow** | **NGS** | **MALDI ToF Mass Spectrometry [13]** |
|---|---|---|---|
| Sensitivity | $10^{-5}$ | $10^{-6}$ | Not defined—Method under development |
| Diagnosis sample needed | No | Yes | Preferred |
| Testing duration | Rapid | 5–10 days | 2 h |
| Standardization | Easy | Mostly service based | Traceable to DA470K |
| Bioinformatics experience needed | No | Yes | No |
| Sample type | Bone marrow | Bone marrow | Serum/Plasma/Urine |
| Cost | Acceptable | High | Acceptable |

## 2.2. PET-CT

Given the heterogeneity and patchy nature of MM, MRD detection in BM might lead to false negative results because of the underestimation of the disease burden due to the presence of the remaining, extra-medullary tumor cells. Because MM is a multifocal disease and residual disease can be present even when BM samples are MRD negative, there is a need for techniques that can detect residual disease outside the BM cavity. Positron-emission tomography computed tomography (PET-CT) imaging using 18F-fluorodeoxyglucose (FDG) as a tracer is the established way of demonstrating such residual disease and can detect disease both inside and outside the BM and has been able to predict patient outcome in myeloma [14,15]. Conventional MRI is of limited use in the evaluation of treatment response [16]. In the future, diffusion weighted MRI might have a role, but this is presently unclear.

Prospective studies have demonstrated that a negative PET-CT provides added value to examination for residual plasma cells in the BM. The data from 192 MM patients treated with induction chemotherapy followed by ASCT demonstrated that persistent PET-CT positive lesions were predictive of shorter PFS (47% vs. 32%) and OS (79% vs. 66%) [17]. These data emphasize the necessity of residual disease evaluation, not only in BM but also outside the BM using sensitive imaging techniques. A combination of MRD assessment in the BM using NGF or NGS and PET-CT evaluation of extramedullary disease can provide additional information on prognosis for MM patients after treatment.

## 2.3. Mass Spectrometry

A novel and highly sensitive method for monitoring the M-component in serum from myeloma patients is under development. The method, named Quantitative Immunoprecipitation Mass

Spectrometry (QIP-MS), uses a polyclonal antibody-based technology to identify and quantify intact immunoglobulins in serum. The technology discriminates proteins based on their molecular mass. Each M-protein is characterized by a specific sequence of amino acids determined during somatic recombination and unique to each plasma cell clone, allowing single-clone tracking over time with high analytical sensitivity. QIP-MS provides a reproducible and sensitive alternative to conventional electrophoresis of serum samples from MM patients [18].

MRD assessments of BM samples involve an invasive method for collection of samples, and alternative methods based on serum are warranted. Ongoing research is exploring QIP-MS as a method of choice for MRD evaluation in MM patients [19]. The question that remains is whether mass spectrometry can fulfill a sensitivity level criterion compared to Euroflow or NGS, which will be required for a serum-based MRD evaluation method. The methods will probably be complementary because the entity being measured is different. Whether mass spectrometry can be used for assessment of MRD in non-secretory MM remains to be seen, and further research is required to document the place for mass spectrometry in this patient population.

## 3. MRD Trials

*Delayed ASCT in NDMM Patients Who Are MRD- and MRD as Treatment Guidance in RRMM Patients*

MM is a heterogeneous disease, which suggests that "one treatment fits all" does not apply. Some patients do not benefit even from the most efficacious therapies available, while others achieve better treatment responses reflected in longer PFS and OS with less efficacious therapies. The questions that remain to be answered are whether MRD status after induction therapy can serve to guide the decision for early versus late ASCT in NDMM patients and if MRD can be used to stop treatment in patients who have achieved a deep and sustained response after first line (1.L) therapy.

An ongoing phase II study in NDMM patients (the MASTER study, NCT03224507) is exploring whether treatment can be stopped when patients achieve MRD negativity. All patients will receive induction therapy consisting of daratumumab (Dara), carfilzomib (K), lenalidomide (R), and dexamethasone (d) followed by ASCT and consolidation with Dara-KRd. Patients who become MRD negative will discontinue therapy (no maintenance therapy) and be actively monitored for recurrence of MRD or clinical relapse. Stopping therapy in MM patients achieving sustained MRD negativity has not been implemented in current guidelines because solid evidence demonstrating long-lasting clinical effects are still lacking. Clinical trials exploring stopping treatment in MRD negative patients will collect data on how quality-of-life and toxicity are affected compared to patients who will be on continuous therapy until progressive disease. For RRMM patients, MRD status might be used to decrease the intensity of treatment, meaning that patients can switch from combination therapy to maintenance therapy if MRD negativity has been reached. This remains, however, to be demonstrated in randomized studies.

The current recommendation of the IMWG is not to treat MM relapse until the criteria of clinical relapse (CRAB symptoms) or biochemical relapse (BR) are met, with no prospective data distinguishing between the two. Several studies are challenging this recommendation, and future guidelines will be based on results from such trials.

An analysis of MM patients who experienced a duration of CR > 24 months prior to relapse after 1.L therapy [20] revealed that starting relapse treatment at biochemical relapse was superior to waiting until symptoms appeared (median OS 125 vs. 81 months). This supports the theory that early treatment at relapse may improve outcome for MM patients. However, randomized trials are needed to verify this.

Interestingly, two small studies indicated that MRD reappearance in previously MRD negative patients precedes biochemical relapse by four months and clinical relapse by nine months [21,22]. During this time period, resistant clones may develop, and tumor burden will increase, making relapse treatment less efficacious.

The ongoing PREDATOR phase 3 study (NCT03697655) is investigating whether preemptive therapy with daratumumab (DARA) can delay clinical relapse in patients with biochemical relapse or MRD reappearance. The PREDATOR study is composed of two phase 2 randomized sub-studies: PREDATOR-BR and PREDATOR-MRD. The former will investigate treatment with DARA in the setting of biochemical relapse, while the latter will test the drug efficacy in patients with MRD reappearance. The PREDATOR-BR sub-study will compare DARA or observation in patients who have achieved at least partial response (PR) to the last line of therapy and experienced asymptomatic biochemical progression. The PREDATOR-MRD sub-study will include patients who have achieved complete remission (CR) or very good partial remission (VGPR) with MRD negativity to the last line of therapy. MRD will be tested by flow cytometry at four-month intervals, and treatment will start at MRD relapse. Patients will be randomized to DARA or no intervention. Results from these studies, and the REMNANT study described in this paper, may generate data supporting early relapse treatment and hopefully improve outcome for MM patients.

MRD-based treatment decision studies can define the depth of response required for sustained benefit, avoid overtreatment of those who have achieved maximal benefit, and clarify whether the drug combination that induces MRD negative state matters beyond the response level itself. In order to use MRD for treatment guidance in MM, robust and reproducible assays with an acceptable cost have to be implemented in clinical practice.

## 4. Early Treatment in Patients with SMM

*Treatment of Patients with SMM*

Smoldering multiple myeloma (SMM) is an intermediate clinical stage between monoclonal gammopathy of undetermined significance (MGUS) and symptomatic myeloma. The standard of care for all patients with SMM has been observation. Recent data have challenged this norm and proposed early intervention as an effective strategy to reduce risk of progression to MM in high-risk (HR) SMM. The paradigm of starting treatment early in MM, that is, in HR-SMM, can serve as an analogue to starting treatment early at first relapse.

The hypothesis that early intervention can reduce the rate of progression in SMM and improve overall survival has been evaluated by the Spanish myeloma group in a phase III study [23] demonstrating that early intervention with lenalidomide and dexamethasone (Rd) led to lower rate of progression to myeloma (39% progression with Rd versus 86% with no intervention) and longer overall survival (18% had died in the Rd group versus 36% with no intervention, median follow-up 75 months) (See Figure 3, below). The trial included patients who had received a diagnosis of SMM within the previous five years and who were at high risk for progression to symptomatic disease. In this trial, high-risk disease was defined as plasma-cell bone marrow infiltration of at least 10% and a monoclonal component (defined as an IgG level of ≥3 g per deciliter, an IgA level of ≥2 g per deciliter, or a urinary Bence Jones protein level of >1 g per 24 h) or only one of the two criteria described above, plus at least 95% phenotypically aberrant plasma cells in the bone marrow plasma-cell compartment, with reductions in one or two uninvolved immunoglobulins of more than 25%, as compared with normal values.

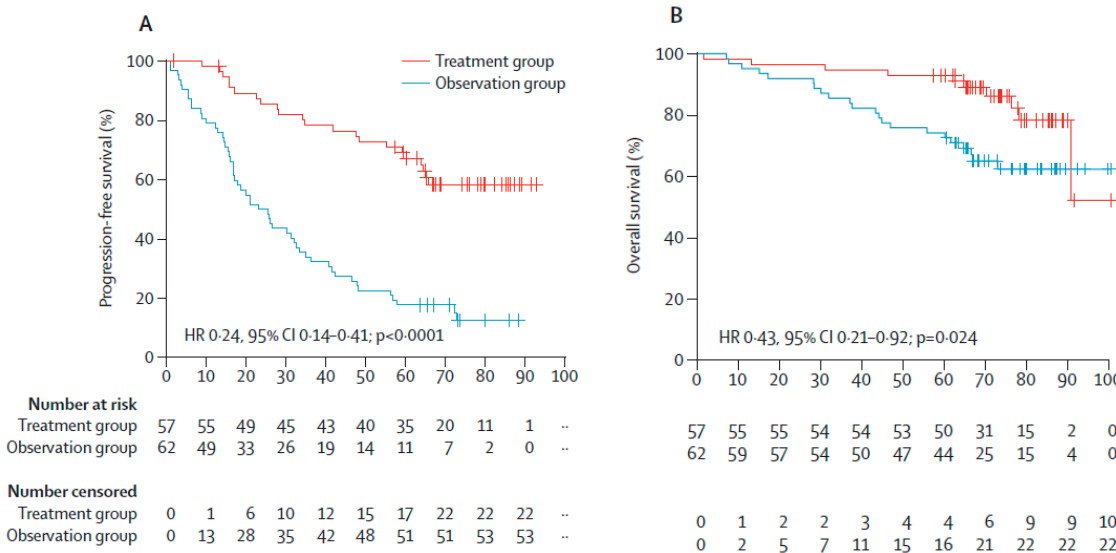

**Figure 3.** (**A**) Progression free survival. (**B**) Overall survival from the point of progression to myeloma. Copyright 2020 with the permission from Elsevier. Reprinted from Mateos et al. [23]. Lenalidomide plus dexamethasone versus observation in patients with high-risk smouldering multiple myeloma (QuiRedex): long-term follow-up of a randomised, controlled, phase 3 trial.

Because the definition of SMM had changed at the time of enrollment, some of the included patients may have had myeloma according to current guidelines, and this might have affected the outcome data.

In a small phase II study, Korde et al. [24] investigated whether treatment of HR-SMM patients with carfilzomib, lenalidomide, and dexamethasone (KRd) could reduce risk of progression. All patients received eight cycles of KRd, and those reaching stable disease (SD) or better continued with two years of lenalidomide maintenance therapy. Nine out of twelve patients (75%) became MRD negative after KRd induction therapy. No patients with SMM experienced disease progression while participating in the study, and the adverse events were manageable. This trial was a phase 2 study with no direct comparison to observation only.

The phase 3 study by Lonial et al. [25] demonstrated that early intervention in patients with intermediate or HR-SMM delayed progression to MM and the development of end-organ damage. PFS was significantly longer with lenalidomide compared to observation with a three-year PFS of 91% versus 66%, respectively. Two deaths were reported in the lenalidomide arm and four were reported in the observation arm.

Recently, a meta-analysis was published by Zhao et al. [26] including eight randomized studies involving 885 SMM patients. Because HR-SMM patients are more vulnerable to progression to MM, a subgroup analysis of two studies enrolling HR-SMM patients was conducted. Patients were treated with Rd versus observation [23] or with thalidomide plus zoledronic acid versus zoledronic acid alone [27]. Both progression and mortality were significantly suppressed by early treatment in comparison to deferred treatment. The meta-analysis indicated that early treatment could significantly slow progression of all SMM patients, and this remained significant regardless of whether the studies enrolling HR-SMM patients were included or excluded.

## 5. Relapse from MRD Negativity as Indication for Treatment: The REMNANT Study

With data supporting the benefit of early treatment in HR-SMM when tumor burden is low, and the established prognostic significance of reaching MRD negativity, the question is; do myeloma patients benefit from earlier relapse treatment?

Here we briefly describe the design and rationale for a prospective, multicenter, randomized, open-label, phase II/III study (ClinicalTrial.gov; NCT04513639) designed to evaluate if treating

measurable residual disease relapse after 1.L treatment prolongs progression free and overall survival for myeloma patients versus treating relapse after 1.L treatment according to the criteria for Progressive Disease (PD) (see Figure 4). To achieve a homogenous group of MRD negative patients, the study consists of two parts. A phase II part with first line treatment in which newly diagnosed patients receive VRd induction and consolidation, with single or tandem autologous stem cell transplant (ASCT). MRD negative patients are then randomized to start treatment at loss of MRD negativity or at progressive disease. The relapse treatment will be daratumumab, carfilzomib, and dexamethasone (Dara and Kd). A total of 391 patients will be included under the assumption that the underlying proportion becoming MRD negative is 45%. This sample size also provides part 2 of the REMNANT study with the required 176 MRD-negative patients under the current assumptions. Should the assumptions be violated, the study will continue until 176 MRD-negative patients have been reached.

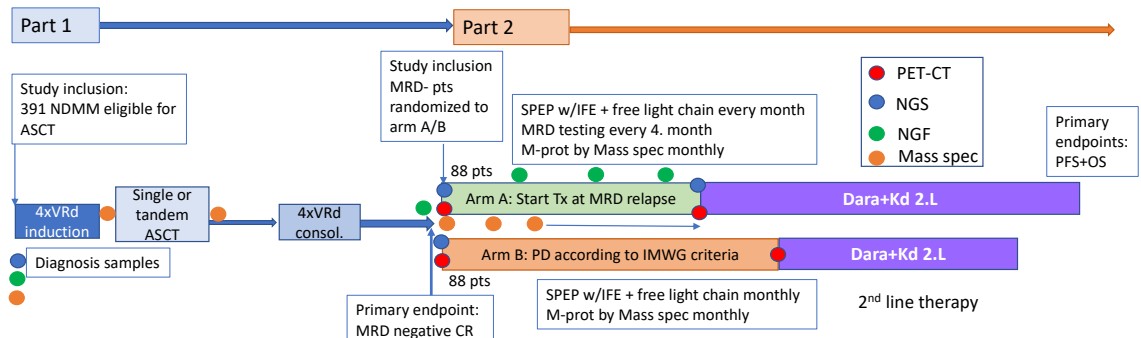

**Figure 4.** Study design. Abbreviations: VRd: Bortezomib, lenalidomide, dexamethasone; NDMM: Newly diagnosed multiple myeloma; ASCT: Autologous stem cell transplant; MRD: Minimal residual disease; PET-CT: Positron emission tomography–computed tomography; NGS: Next generation sequencing; NGF: Next generation flow; Mass.spec: Mass spectrometry; Dara+Kd: Daratumumab, carfilzomib, dexamethasone.

## 5.1. First Line Treatment

The three-drug combination of bortezomib, lenalidomide, and dexamethasone has demonstrated deep responses in NDMM [28–31], resulting in MRD negative rates of 30–60%, dependent on the sensitivity rate of the assay used for assessment of MRD [1,31,32]. In the transplant eligible population, the IFM 2009 trial demonstrated that induction therapy with three cycles of VRd, immediate ASCT, two cycles of VRd consolidation, and 12 months of lenalidomide maintenance is superior compared to eight cycles of VRd therapy followed by 12 months of lenalidomide maintenance in terms of MRD negativity rates, MRD $10^{-4}$ rates of 63% vs. 49%, MRD $10^{-6}$ rates of 30% vs. 20%, and median PFS; 50 months vs. 36 months (HR 0.65%, 95% CI 0.53–0.8; *p*-value < 0.001), favoring the transplant group [1,30]. Today, the VRd induction regimen is an accepted standard-of-care, used as a control arm in most ongoing studies in this population, although not approved by the European Medicines Agency (EMA). In our study, we have chosen VRd for induction because this is the best available regimen in Norway. In a recently published retrospective study, Blocka et al. analyzed 978 trial and non-trial patients who underwent single or tandem ACST across Germany. The result showed that patients who had an improved response after ASCT benefited significantly from the tandem ASCT with increased PFS compared to patients without improved response status [33]. Recent results of the EMN02/HO95 trial, a phase III randomized multicenter study, demonstrated a clinical benefit in terms of PFS (median: 47 vs. 38 months) and OS (estimated 10-yr probability: 58% vs. 47%; HR 0.69, CI 0.56–0.84, *p* = 0.0002) of tandem ASCT vs. single ASCT [34]. Tandem ASCT will, for these reasons, be given to patients eligible and willing to perform a second transplant. Low dose lenalidomide monotherapy has been shown to improve OS in myeloma patients when used as maintenance therapy after HDM-ASCT [35], but is not reimbursed in Norway. If lenalidomide maintenance will one day become reimbursable, the study will be amended and patients in both arms will receive this treatment after consolidation therapy.

The primary objective of the phase II study is to determine the MRD negativity rate after Norwegian standard of care (SOC) 1.L treatment in NDMM patients eligible for ASCT. Secondary objectives include PFS, OS, and overall response rate (ORR) after Norwegian SOC 1.L treatment, the rate of sustained MRD negativity, and safety. Exploratory objectives are the impact on health related quality of life (HRQOL) of Norwegian SOC 1.L treatment; to quantify serum monoclonal immunoglobulin during 1.L treatment using QIP-MS; and the differences in MRD negativity after induction, ASCT, and consolidation in a predetermined subpopulation.

*5.2. Second Line Treatment*

Patients in the REMNANT study will receive second line treatment upon loss of MRD negativity in arm A, and at PD in arm B. MM patients who are exposed or refractory to either lenalidomide or bortezomib, or both, have limited options for regiments that are both efficacious and tolerable [36]. Both carfilzomib and daratumumab (SC and IV), in combination with lenalidomide or dexamethasone, or both, are approved by the EMA at first relapse, and daratumumab is also approved in combination with bortezomib and dexamethasone. A recent phase 1b study evaluated the combination of daratumumab, carfilzomib, and dexamethasone (Dara and Kd) in 85 MM patients who had received at least one prior line of therapy, including patients exposed to both bortezomib and lenalidomide. All patients had received prior treatment with bortezomib and 95% had received prior treatment with lenalidomide. The conclusion was that the combination was well-tolerated with a safety profile consistent with the safety profile of the individual agents and that the response was encouraging. At a median follow-up of 16.6 months, the ORR was 84%, >VGPR rate of 71% and >CR rate of 33%. Median PFS was not reached in the intention to treat (ITT) population, with 12- and 18-month PFS rates of 74% and 66%, respectively. The 12-month OS rate was 82% [37].

Preliminary results from the ongoing phase III study (CANDOR trial) of carfilzomib, dexamethasone, and daratumumab (DKd) vs. carfilzomib and dexamethasone (Kd), including 466 RRMM patients, has demonstrated a significant benefit in terms of PFS for the Dara and Kd group, where median PFS was not reached vs. 15.8 months in the Kd group (HR, 0.63; CI, 0.46–0.85; $p = 0.0014$). Importantly, Dara and Kd was also significantly beneficial in the lenalidomide exposed/refractory groups, with a median PFS not reached vs. 12.1 months in the lenalidomide exposed group (HR, 0.52; 95% CI, 0.34–0.80) and median PFS not reached vs. 11.1 months in the lenalidomide refractory group (HR, 0.45%; 95% CI, 0.28–0.74) [38]. Weekly dosing of carfilzomib 20/70 mg/m$^2$ has been demonstrated to be safe in combination with daratumumab [37] and to significantly increase PFS (11.2 months vs. 7.6 months) compared to twice weekly in combination with dexamethasone [39]. Based on these results, Dara and Kd was chosen as the most appropriate regimen for relapse treatment in this study. The primary objectives of the phase III study are to demonstrate the benefit of initiating 2.L treatment at loss of MRD negativity compared to initiating 2.L treatment at progressive disease according to IMWG criteria. Secondary objectives are to determine the time from randomization to start of 3.L treatment time to next treatment, to determine the rate of MRD negativity during 2.L treatment, to evaluate how starting treatment early vs. late impacts HRQOL, and to evaluate safety. Exploratory objectives include comparing depth of response between NGS and Euroflow NGF, exploring the correspondence between PET-CT results and MRD dynamics (Euroflow NGF, NGS) by intra-patient comparison, and comparing bone marrow MRD negativity with QIP-MS.

## 6. Conclusions

MRD negativity has become an important prognostic tool, but has yet to evolve to a predictive tool that can determine treatment decisions. Many studies are underway to decide what to do with patients accomplishing or not accomplishing MRD negativity after 1.L treatment. The novelty of this study is to evaluate what to do at loss of MRD negativity, and hopefully to prove the benefits of early relapse treatment.

**Author Contributions:** Conceptualization, A.-M.R., F.B.A., and F.S.; project administration, A.-M.R.; writing—review and editing, A.-M.R., F.B.A. and F.S.; supervision, F.S.; funding acquisition, A.-M.R. and F.S. All authors have read and agreed to the published version of the manuscript.

**Funding:** This research was funded by the Norwegian government, the Norwegian Cancer society, Celgene, Janssen, and The Binding Site.

**Acknowledgments:** We would like to acknowledge and thank our sponsors, which include the Norwegian government, the Norwegian Cancer society, Celgene, Janssen, and The Binding Site. We thank Tobias Schmidt Slørdahl for critically reading the manuscript.

**Conflicts of Interest:** The funders had no role in the design of the study; in the collection, analyses, or interpretation of data; in the writing of the manuscript, or in the decision to publish the results.

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
