# Peer review of "The Next Step for MRD in Myeloma? Treating MRD Relapse after First Line Treatment in the REMNANT Study"

_hemato, doi:10.3390/hemato1020008_

Round 1

Reviewer 1 Report

The article of Rasmussen et al. examines what they think to be the next step for MRD in MM: treating MRD relapse in their so-called REMNANT study, believed to start very soon. The manuscript is split into 4 sections: (i) an introduction on MRD in MM focusing on the recent meta-analyses; a so-called “materials and methods” section, which is rather the list of the techniques used to monitor MRD; a so-called “results” section, which include data from the literature supporting the REMNANT trial; and a last section particularly detailed about study design.

This article format is very unusual but not uninteresting. Nevertheless, I would recommend to synthesize more certain parts, whose detail seems excessive in the context of an article: perhaps the complete design of the study has no place here? If readers want the details, they could be redirected to the link https://clinicaltrials.gov/ct2/show/NCT04513639. Rather, the purpose of the article is to discuss how MRD is changing our practices, now and in the future, and to explain what major questions this trial will help answer.  

Minors issues :

-          Line 19 : typing mistake “om MRD”

-          Figures 1-3: low definition quality, source not detailed, not referred in the text, no legend.

-          Lines 72-73-74: please refer to Prentice RL. Surrogate endpoints in clinical trials: definition and operational criteria. Stat Med 1989; 8:431-40.

-          Line 128: the authors claim that sustained MRD negativity has been correlated with better outcome: a reference should be done to the reader, or the authors should reword this sentence.

-          Line 129: the concept of PET-CT (that is addressed for the first time in the manuscript) should be dissociated from the concept of sustained MRD for more clarity.

-          PET-CT section: a short comment about lack of MRI specificity should be included.

-          Line 194: compared to MRD performed in the BM whatever the used technique (why  EuroFlow specifically ?)

-          Ref 20: typing mistake

-          Line 246: typing mistake (a MRD negative state)

-          In the chapter “Early treatment in patients with SMM”: the authors should clarify if the patients enrolled in these trials were ex-SMM (MM with slim criteria) or not, and how were defined HR SMM ?

-          Line 308: typing mistakes

-          Lines 332, 340: typing mistake (patients, not pts)

Author Response

REVIEWER 1 SPECIFIC POINTS
Minor issues:
The heading “Materials and Methods” has been changed to “Techniques for MRD monitoring”.
Line 19, typing mistake has been corrected.
Figures 1-3: We have improved quality of the figures and figure text have been added, including copy rights.
Lines 72-73-74: A new reference to RL Prentice has been included as suggested by the reviewer.
Line 128: We have included a meta-analysis by Munshi et al. JAMA Oncol. 2017 as a new reference.
Line 129: We would like to inform that PET-CT cannot be dissociated from the concept of sustained MRD, because the "Sustained MRD-negative" response category includes MRD negativity by imaging with PET. Therefore, no changes have been made to the manuscript.
PET-CT section: We have added a sentence commenting on MRI and a reference by Hillengass Lancet Oncol. 2019.
Line 194: NGS has been added to the text.
Ref. 20 has been corrected.
Line 246: Has been corrected to “a MRD negative state”.
We agree with the reviewer that the data presented from the SMM trial by Mateos and coworkers might have been influenced by the previous definition of SMM. We have therefore added more details about the patient population included in the trial and definition of SMM at time of inclusion in Chapter “Early treatment in patients with SMM”.
Line 329: The heading “Study design” has been changed to “Relapse from MRD negativity as indication for treatment: The REMNANT study”
Line 334: “Briefly” has been added to the text.
Line 338: “After treatment with standard-of-care first line treatment” has been removed from the text.
Line 343: Typing mistake has been corrected.
Line 359: Typing mistake has been corrected.
Line 389: Typing mistake has been corrected.
Line 390: Typing mistakes has been corrected.
Line 399: Typing mistake has been corrected.
Major issues:
We agree with the reviewer that the complete design of the study is too much detailed and has therefore removed sections from this part of the article.
Line 418 to 424: The section “Objectives of the study part one” has been moved to the section “First line treatment”.
Line 425 to 433: The section “Objectives of the study part one” has been moved to the section “Second line treatment”.
Line 434: The box “Key eligibility criteria” has been removed from the text.
The section “Study location” has been removed from the text.
The section “Study design and methodology” has been removed from the text.
The section “Outcome measurements/endpoints” has been removed from the text.
The section “Statistical analyses” has been removed from the text.
Line 518: Part one of the section “Sample size” has been moved to line 342 under the section “Relapse from MRD negativity as indication for treatment: The REMNANT study”. The rest of the section has been removed from the text.

Reviewer 2 Report

The review is well written, concise and interesting. Few important things should be improved.

The quality of figures is not good, this should be improved. In the text, there should be reference to the figures and each figure should have a clear legend describing what is on the figure, from which clinical trial etc.

I suggest to include a table in the section of Material and Methods comparing pros and cons of each MRD assessment method,  sample that is required, sensitivity that is reached etc.

Mass spectrometry based MRD-assessment: this method however cannot properly assess MRD if MM clones are non-secretory. How is it, if it sectretes only light chains? Few sentences here should be mentioned or in a table describing pros and cons of the methods.

Overall, few sentences should be addedd to explain the benefit of switch of the therapy or stop of the teratment in case patient reaches MRD negativity - such as less toxicity for the patient, but also potentially less pressure on the cells to select for or evolve into highly therapy resistant clone.

Author Response

REVIEWER 2 SPECIFIC POINTS
Figures 1-3: We have improved quality of the figures and figure text have been added, including copy rights.
We agree with the reviewer that a table will clarify the pros and cons of the different MRD assessment methods and we have therefore included a table under “How to monitor MRD”.
Mass spectrometry: We have added a sentence about the use of the technique in patients with non-secretory MM.
A sentence about stopping therapy for patients achieving MRD negativity has been added.